# Effects of Different Moisture-Permeable Packaging on the Quality of Aging Beef Compared with Wet Aging and Dry Aging

**DOI:** 10.3390/foods9050649

**Published:** 2020-05-18

**Authors:** Yingwu Shi, Wangang Zhang, Guanghong Zhou

**Affiliations:** Key Laboratory of Meat Processing and Quality Control (MOE), Key Laboratory of Meat Processing (MOA), Jiangsu Synergetic Innovation Center of Meat Processing and Quality Control, Nanjing Agricultural University, Nanjing 210095, China; 2017116006@njau.edu.cn (Y.S.); ghzhou@njau.edu.cn (G.Z.)

**Keywords:** beef, aging, moisture-permeable packaging, quality

## Abstract

The objective of this study was to investigate the effects of six different aging methods (four types of moisture-permeable packaging, wet aging, and dry aging) and aging time (0, 7, and 14 d) on the quality of aging beef, especially physicochemical properties. The weight loss, aerobic bacterial counts, yeast counts, and mold counts increased with the increase of moisture permeability and aging time. However, shear force, hardness, cohesiveness, and chewiness followed an opposite trend with increasing moisture permeability. The values of L* and b* appeared to decrease in the dry-aged samples compared with those of the others. In addition, water content in dry-aged samples for 7 and 14 d showed a significant decrease. The higher myofibril fragmentation index was observed in dry-aged samples for 7 and 14 d compared with groups using moisture-permeable packaging. Meanwhile, the percentage of bound water and free water appeared to decrease with the increase of moisture permeability. Thus, different moisture-permeable packaging was able to control different levels of water loss and effectively reduce microbial contamination compared with dry aging. The changes of both myofibrillar fragmentation index (MFI) and distribution of water indicated that moisture-permeable packaging affected the structure of myofibrils, which influenced the shear force.

## 1. Introduction

The aging of beef, especially dry aging, plays an important role in increasing of tenderness and the improving of flavor [1,2]. Dry aging and wet aging, as two main traditional aging approaches, have been widely used in beef production in recent decades. However, these methods have their own advantages and disadvantages. Wet aging has more stable quality and a higher yield compared to dry aging, which has more desirable palatability [3]. Excessive weight losses and risky microbial contamination from dry aging make practical production unfavorable, while a lack of sufficient flavor appears in wet-aged beef. Thus, some researchers are aware of the feasibility of applying moisture-permeable packaging to package raw beef, so that packaged beef is similar to dry-aged beef in terms of the palatability attributes.

Moisture-permeable packaging is made of thermoplastic urethane. Some functions of moisture-permeable packaging, such as high softness, high elasticity, cold resistance, and anti-mildew resistance, play a positive role in the process of simulating dry aging. Moreover, no other functions of these moisture-permeable packaging are more important than moisture penetrability. Moisture-permeable packaging is a kind of low-price, environmentally friendly, nontoxic, and degradable polymer film that is food grade. It has great application prospects in the process of aging beef. Ahnstrom [4] found that the beef packaged with moisture-permeable packaging had the same flavor and taste compared with the traditional dry-aged beef. This moisture-permeable packaging was not only able to reduce weight loss, trim loss, and microbial contamination but could also increase the yield of beef. Li [5] found that the flavor of beef packaged with moisture-permeable packaging was much more tender, juicier, and more enjoyable than that of wet-aged beef, although more weight loss occurred. Likewise, the study from Stenstrom [6] showed that consumers tended to choose beef that was dry aged or aged in moisture-permeable packaging rather than wet aged.

In general, the studies mentioned above reported the application of different moisture-permeable packagings, such as those having moisture permeabilities of 2500, 5000, and 8000 g/m^2^/24 h. Single packaging was explored in the existing study rather than the comparison of several packagings. The difference of moisture permeabilities may result in various consequences such as the difference of microbial population and others. In addition, there are contradictions in the existing research. The study of Dikeman [7] showed that moisture-permeable packaging had no advantages over wet aging. Parrish [8] and Sitz [9] studied the influence of dry aging and wet aging on tenderness and found no difference between the two. Nevertheless, Kim [10] found that dry-aged beef at 1 °C resulted in a lower shear force than that of wet-aged beef under the same environment. Therefore, the purpose of this study was to investigate the effects of six different aging methods (four kinds of moisture-permeable packaging, wet aging, and dry aging) and aging times (0, 7, and 14 d) on the quality of aged beef, especially its physicochemical properties.

## 2. Materials and Methods

### 2.1. Sample Preparation

Twelve crossbred Luxi bulls (approximately 2 years old) were slaughtered at a slaughter company (Shandong Hongan Co. Ltd., Binzhou, China) according to the Operating Procedures of Cattle Slaughter in the National Standards of PR China. Longissimus thoracis (LT) muscles were obtained from both sides of each carcass at 48 h post-slaughter. Every LT was divided into three parts on average, and all of them were randomly assigned to six aging methods (wet aging, four kinds of moisture-permeable packaging, and dry aging). All samples were vacuum packaged, stored in a plastic foam box with ice and transported from the company to the laboratory. Then, LT sections for wet aging were vacuum packaged with packaging solutions (model DCS00-FB-E; PROMAX, Rancho Cucamonga, CA, USA) in vacuum bags having an O_2_ permeability of 74 mL/m^2^/24 h at 4 °C, a water vapor permeability of 3.8 g/m^2^/24 h at 38 °C, and 90% relative humidity. LT sections without packaging (dry aging) were placed in a refrigerator room at 2 °C. LT sections assigned to moisture-permeable packaging were vacuum packaged in moisture-permeable packaging (320 × 480 × 0.03 mm^3^; water vapor transmission rates of 350, 5000, 7900, and 11,000 g/m^2^/24 h at 23 °C and 50% relative humidity, respectively; O_2_ permeability of 1928, 2323, 2265, and 3010 mL/m^2^/24 h at 4 °C respectively; Dongguan Dingzheng Environmental Protection Material Co. Ltd., Dongguan, China). All samples were kept on the stainless shelves with an air flow of 1.5 m/s, a temperature of 2 °C and a relative humidity of 85% for 0, 7, and 14 d. The weight of all samples was recorded before aging, after aging, and after trimming. After aging for 0, 7, and 14 d, respectively, the samples were trimmed, and the pH, color, and other indexes were measured. The remaining samples were vacuum packaged and stored at −20 °C for 30 d. 

### 2.2. Weight Losses

Weight was recorded before aging, after aging, after trimming any dry and discolored portions of the beef, and after trimming connective tissue, respectively. The percentage of aging loss was calculated as:% aging loss = ((weight before aging − weight after aging)/weight before aging) × 100%(1)

The percentage of trimming loss was calculated as: % trimming loss = ((weight after aging − weight after trimming dry and discolored parts)/weight after aging) × 100%(2)

The percentage of total loss was calculated as: % total loss = ((weight before aging − weight after trimming connective tissue)/weight before aging) × 100%(3)

### 2.3. Ph, Color, and Cooking Loss

The pH of all the samples was measured manually after aging, using a portable pH meter (HI9025, Hanna Co., Villafranca Padovana, Italy). The pH meter was calibrated with pH 4.0 and 7.0 standard buffers before measurement. The pH values were recorded after the number stabilized. Every sample was measured in three different locations, and an average value was obtained for statistical analysis. The color was measured on a section of every piece of beef with a colorimeter (CR-40, Minolta Camera, Tokyo, Japan). Every sample was cut into 2 × 3 × 5 cm^3^ sections and the weight of these was recorded before cooking and after cooking. These samples were cooked at 72 °C in a water bath kettle until the center temperature reached 70 °C. The center temperature of each beef cube was monitored using a digital thermometer (SSN11E, YUWESE, Shenzhen, China).

### 2.4. Shear Force and Texture Profile Analysis (TPA)

After the beef was cooled under running water for 30 min and dried with paper towels, samples were cut parallel to the muscle fibers with a cross-sectional area of 1 × 1 cm^2^. Shear force was measured by using an apparatus (model XL1155, Xielikeji. Co., Ltd., Qinghuangdao, China). The shear velocity was 5 mm/s. The sample size was a 1 × 1 × 1 cm^3^ cuboid for the physical property tester (TA XT Plus, Stable Micro Systems, Surrey, UK). The height of the measurement was 30 cm, and the compression ratio was 75% with a pretest speed, measuring speed, and posttest speed of 5 mm/s. Hardness, springiness, cohesiveness, and chewiness were measured.

### 2.5. Water Content

Two grams of meat was removed from the samples and put into a 10 mL beaker that had been dried. The small beakers with two grams of samples were kept in an oven (DGG-9240A, SUMSUNG., Co., Ltd., Suzhou, China) at 105 °C for 12 h. The water content was calculated as: Water content = ((original weight − weight after being dried)/original weight) ×100(4)

### 2.6. Microbiological Analysis

Aerobic bacterial count, lactic acid bacteria, and yeast and mold populations were enumerated by aseptically removing 25 g of samples from the inner layer after trimming. These samples were placed into Stomacher blender bags mixed with 225 mL bacteria-free physiological saline (0.85%), and mixed in a blender (BagMixer400; Interscience, Lyon, France) for 1 min. The proper serial decimal dilutions of the homogenate were conducted with bacteria-free physiological saline (0.85%). The appropriate dilutions of 1 mL were inoculated into Petri dishes mixed with the corresponding medium. Plate count agar (Dalian Meilun Biological Technology Co., Ltd., Dalian, China) was incubated at 37 °C for 48 h in a biochemical incubator (Shanghai Boxun Industry and Commerce Co., Ltd., Shanghai, China) to determine aerobic bacterial counts. De Man, Rogosa, and Sharpe agar (MRS agar) (Dalian Meilun Biological Technology Co., Ltd., Dalian, China) was incubated at 37 °C for 72 h to determine lactic acid bacteria counts. Potato dextrose agar (Dalian Meilun Biological Technology Co., Ltd., Dalian, China) was incubated at 25 °C for 120 h in a biochemical incubator to determine yeast and mold counts. Incubation for lactic acid bacteria was implemented under an anaerobic environment. Counting the amount of bacterial colonies in each plate was performed with a bacterial colony counter (Scan 1200, Interscience, France). The results are expressed as log colony-forming unit (cfu)/g meat.

### 2.7. Myofibrillar Fragmentation Index (MFI)

MFI was determined according to Xiong [11] with slight modifications. The beef samples (4 g) were homogenized using a high-speed homogenizer (T10 basic, Ika Company, Staufen, Germany) at 8000 rpm for 30 s with 40 mL of homogenate buffer (including 100 mmol/L KCl, 20 mmol/L K_2_HPO_4_, 20 mmol/L KH_2_PO_4_, and 1 mmol/L EGTA, pH 7.0). The homogenate buffer was kept in a beaker (4 °C). The solution was centrifuged at 5000× *g* at 4 °C for 15 min, and the supernatant was discarded. The above operation was repeated to obtain the precipitation. The precipitate was washed with 20 mL of homogenate buffer and mixed fully. The solution was filtered with one layer of gauze to remove the connective tissue. The protein solution was centrifuged at 5000× *g* at 4 °C for 15 min. The supernatant was discarded and dissolved with 8 mL of homogenate buffer. The protein concentration was measured by using the bicinchoninic acid (BCA) protein assay kit (P0012S, Gensmart, Nanjing, China) and a microplate reader (SpectraMax M3, Molecular Devices Company, San Francisco, CA, USA) at 540 nm. The protein concentration was diluted to 0.5 mg/mL. The absorbance value was multiplied by 200 to obtain the MFI.

### 2.8. Low-Field NMR Analysis

The distribution of water was measured by a Niumag benchtop pulsed NMR analyzer (PQ001, Niumag Electric Corporation, Shanghai, China). Two grams of samples were placed in an NMR tube with a diameter of 15 mm. Relaxation time (T_2_) was measured using the Carr–Purcell–Meiboom–Gill (CPMG) sequence. Measurement parameters included repetition time (RT) = 4500 ms, spectral width (SW) = 100 KHz, number of slices (NS) = 8, and number of echoes (NECH) = 4000. The low-field NMR relaxation T_2_ curve was fitted to a multiexponential curve with the MultiExp Inv Analysis software (Niumag Electric Corporation, Shanghai, China).

### 2.9. Statistical Analysis

The experiment was designed as a randomized complete block with four replications. The treatment structure was a 3 × 6 factorial with six aging methods (dry aging, wet aging, and four different moisture-permeable packagings) and three aging times (0, 7, and 14 d). Statistical analysis was implemented in Statistical Product and Service Solutions (SPSS version 24.0, International Business Machines Co., Armonk, NY, USA). The MIXED procedure was applied with the aging method, aging time, and their interactions as fixed effects and the animals as random effects.

## 3. Results

### 3.1. Weight Losses

As shown in Table 1, there were significant effects of aging methods and aging time on aging loss, trimming loss, and total loss (*p* < 0.05). Aging loss, trimming loss, and total loss increased with the increasing moisture permeability. Aging losses on day 14 were significantly greater than those on day 7 except for the wet aging treatment (*p* < 0.05). The dry aging treatment resulted in the highest aging losses between treatments, which is consistent with previous studies [12,13]. The lowest aging losses appeared in wet-aged samples due to vacuum-packaged bags having the lowest moisture permeability. Aging losses in moisture-permeable packaging were intermediate between those of wet aging and dry aging, which is similar to the results from Li [5], who reported that aging losses of dry-aged samples were higher than those of samples aged in moisture-permeable packaging. Lower moisture permeability inhibited the evaporation of water, so aging loss decreased with the decreasing moisture permeability. Dry and discolored parts of the aged beef were cut due to excessive water loss of the beef surface, so that trim loss was apparently influenced by aging methods and aging time (*p* < 0.05). There was no trim loss in the samples aging in vacuum bags and two low moisture-permeable packagings (350 and 5000 g/m^2^/24 h) for 7 d. It was likely that moisture-permeable packaging with low moisture permeability led to low evaporation of water within 7 d. However, the appearance of trim loss in the samples aging for 14 d indicated that a long enough aging time still led to trim loss despite low moisture permeability. There was an increasing tendency in total loss of treatments apart from wet aging when aging time and moisture permeability of packaging increased, because the vacuum bags had very low moisture permeability. The results above indicated that moisture-permeable packaging was able to effectively control the evaporation of water, thereby moisture-permeable packaging can effectively reduce weight losses compared with dry aging. Similarly, the aging loss, trim loss, and total loss of the samples packaged in moisture-permeable packaging were intermediate between those of the dry-aged and wet-aged samples, which are in accordance with previous studies [4,14,15]. These results confirm that moisture-permeable packaging could control the level of weight loss.

### 3.2. pH, Color, and Cooking Loss

The pH values were relatively stable, ranging from 5.37 to 5.41 on day 0. However, a slight increase in pH on days 7 and 14 was observed for all treatments compared with samples aged for 0 d (Table 2). Protein degradation produces alkaline substances that may contribute to the increase in pH value observed on days 7 and 14. Although aging time influenced pH significantly (*p* < 0.05), aging methods and their interaction were not significantly different (*p* > 0.05). This result showed that water evaporation could not have a significant effect on the pH value. These consequences conflicted with the study of Li [5], who concluded that there a lower pH was observed for wet aging than for dry aging and for aging in moisture-permeable packaging for 8 and 19 d. However, Degeer [15] presented that there was no significant difference between different aging methods.

The L* value was significantly influenced by aging methods rather than by aging time. Aging time had an apparent influence on the a* and b* values, as shown in Table 2 (*p* < 0.05). The L* value was significantly lower in dry-aged samples than in wet-aged samples (*p* < 0.05). An obvious decrease in the b* value was observed in all aging methods apart from wet aging (*p* < 0.05). This could be due to the greater moisture permeability resulting in a lower water content and a darker surface of the samples. In general, there was a decreasing tendency in the L* and b* values when moisture permeability increased. However, the a* value showed no significant difference between aging methods. Higher moisture permeability resulted in higher water loss in the outer layer and then influenced the inner layer of samples, contrasting with the wet-aged samples. Therefore, higher water loss may lead to the occurrence of slight discoloration in the inner layer of samples aged in highly moisture-permeable packaging. However, Li [5] found that there was no significant difference in color between wet aging and dry aging.

As shown in Table 2, the cooking loss from samples wet aged for 14 d was significantly lower than that for those wet aged for 0 or 7 d and for those subjected to other treatments for 14 d (*p* < 0.05). In addition, cooking loss from the samples dry aged for 14 d was higher than that of those dry-aged for 0 and 7 d (*p* < 0.05), while no difference was observed in samples aged in moisture-permeable packaging between 7 and 14 d. However, Warren [16] reported that dry-aged samples had lower cooking loss than unaged samples. The myofibril structure of the dry-aged samples was slightly damaged; therefore, more moisture was released when the beef was cooked. No difference in cooking loss was found among dry aging, wet aging, and aging in moisture-permeable packaging [4,10].

### 3.3. Water Content

The water content of beef averaged 73.6% for 0 d (Table 2). The water content of samples dry aged for 7 and 14 d was apparently lower than that of the samples subjected to other treatments (*p* < 0.05), but no significant difference was found between wet aging and aging in moisture-permeable packaging. Ahnstrom and Lee [4,17] found no significant difference in water content between dry aging and aging in moisture-permeable packaging. However, a significant difference was found among wet aging, dry aging, and aging in moisture-permeable packaging [7,18]. Although dry aging had an apparently higher total loss than the other treatments in our study, the water content of dry-aged samples exhibited a slight decrease and the other treatments had no significant difference. This indicates that the interior of dry-aged beef could still maintain a sufficient water content while drying occurred on the beef surface. This is a possible reason that rigid and dry surface of dry-aged beef was beneficial for preventing further water loss.

### 3.4. Shear Force and Texture Profile Analysis (TPA)

A significant interaction effect was observed on the shear force between aging time and aging methods (*p* < 0.05, Table 3), and aging time significantly influenced shear force (*p* < 0.05). As shown in Table 3, the shear force of dry-aged beef was less than that of other treatments for samples aged for either 7 or 14 d (*p* < 0.05). Moreover, there was a decreasing tendency in the shear force with the increasing of moisture permeability. Sufficient water evaporation might have an influence on the inner structure of beef. Therefore, the myofibril network was damaged at different levels, and the tenderness of beef was improved. The results above are similar to those of the study by Kim [10,19], who found that dry aging imparted a lower shear force on samples than wet aging. However, the results from this study contradict those of Lepper-Blilie [20], who reported that there was no significant difference in tenderness between dry aging and wet aging.

Similarly, the hardness, cohesiveness, and chewiness of beef exhibited similar trends to the shear force (Table 3). There were significant effects of aging time, aging methods, and their interaction on hardness; in particular, dry aging for 14 d imparted the lowest hardness. This decreasing tendency with the increasing of moisture permeability was similar to that of shear force. Cohesiveness reflected the strengths and weaknesses of the binding functions of molecules or structural elements in the samples. Aging time and the interaction of aging time and method had an apparent influence on cohesiveness (*p* < 0.05), while the aging method alone had no significant effect (*p* > 0.05). The reason for the decreasing cohesiveness with the increasing moisture permeability might be the internal myofibrillar structure was damaged leading to decreasing binding force between the internal molecules. Chewiness refers to the energy required for chewing solid samples, which comprehensively reflects the continuous resistance of the samples to chewing [20]. Chewiness was related to hardness and cohesiveness, as chewiness decreased with the decreasing hardness and cohesiveness in our study. Caine [21] also found that there was a significant correlation among shear force, hardness, cohesiveness, and chewiness. Springiness is a mechanical textural attribute related to the rapidity and degree of recovery from a deforming force [22]. Both aging methods and interaction had no significant influence on springiness. Caine [21] also reported that springiness had no significant correlation with shear force and hardness. In general, the results showed that hardness, cohesiveness, and chewiness were related to shear force when they had a similar variation trend.

### 3.5. Microbiological Analysis

Aging time apparently affected the microbial population, especially the aerobic bacterial count and yeast (*p* < 0.05). Significance was evident among treatments for aerobic bacterial count, lactic acid bacteria, yeast, and mold. Increasing amounts of aerobic bacterial count, yeast, and mold with the increasing moisture permeability were observed in Table 4. Samples aged in moisture-permeable packaging had lower aerobic bacterial counts than dry-aged samples (*p* < 0.05), while wet-aged samples had the lowest aerobic bacterial counts. Aerobic bacterial counts became gradually higher with the increase of moisture permeability, but Degeer [15] reported that aerobic bacterial counts from the surface of dry-aged beef were lower than those of beef aged in moisture-permeable packaging. Compared with the dry-aged samples, wet-aged samples were under a vacuum environment, which is able to inhabit the growth of microorganism, thus, these samples had lower aerobic bacterial counts. 

In contrast, the populations of lactic acid bacteria of the wet-aged samples were apparently higher than those of the other treatment samples aged for 14 d (*p* < 0.05), although the lactic acid bacteria count for wet-aged samples after 7 d were lower than those for the other samples subjected to other treatments (*p* < 0.05). Vacuum bags had lower oxygen permeability than the moisture-permeable packaging, so that this environment benefited the growth of lactic acid bacteria, which was anaerobic. Similarly, Li [18] found that the enumeration of lactic acid bacteria in wet aging was larger than that in dry aging for 8 or 19 d. However, treatments apart from wet aging resulted in lower lactic acid bacteria populations after 14 d than those after 7 d. A possible reason was that moisture-permeable packaging had slight oxygen permeability, resulting in a disadvantaged environment with more oxygen, therefore, this environment inhibited the growth of lactic acid bacteria. Similar results were reported by Li [18], who found that lactic acid bacteria counts were higher in vacuum bags than in moisture-permeable packaging. With the increase in moisture permeability, the surface of beef became dryer and harder, so that the hard surface preventing oxygen penetration made lactic acid bacteria counts become higher ranging from treatment with moisture permeabilities of 350 g/m^2^/24 h to dry aging. 

There was no significant difference in the enumeration of mold between treatments aged for 7 d, but higher mold counts were observed for dry-aged samples than for the others after 14 d (*p* < 0.05). It was possible that moisture-permeable packaging was a barrier to external mold. Mold counts of the samples aged in moisture-permeable packaging were lower than those of the dry-aged samples. The mold count of the dry-aged samples for 7 d was 1.34 log cfu/g, and the larger mold counts (3.14 log cfu/g) appeared in dry-aged samples for 14 d. Similar results were obtained from the study of Li [5], who reported that dry-aged samples had an increasing tendency with regard to mold counts with the extension of aging time. The yeast counts tended to increase with the increase of moisture permeability after 7 and 14 d. Yeast counts of the dry-aged samples were higher after 14 d than after 7 d (*p* < 0.05); however, yeast counts of the wet-aged samples were lower for 14 d than for 7 d (*p* < 0.05). Ahnstrom [4] also reported that yeast counts were greater in dry-aged samples than in those aged in moisture-permeable packaging. In general, we hypothesized that moisture-permeable packaging and hard surfaces protected the internal environment of beef and reduced the amount of microorganisms.

### 3.6. Myofibrillar Fragmentation Index (MFI)

MFI reflects the integrity of myofibrillar and skeletal proteins [11]. It is shown in Table 5 that aging methods, aging time, and their interaction had significant effects on the MFI (*p* < 0.05). There was no significant difference in MFI among the six treatments on day 0. However, the MFI increased with increasing moisture permeability after either 7 or 14 d. When beef was aged for 7 d, the groups with moisture permeabilities of 7900 and 11,000 g/m^2^/24 h had a significantly higher MFI than that with moisture permeabilities of 350 g/m^2^/24 h (*p* < 0.05), but the groups with moisture permeabilities of 7900 and 11,000 g/m^2^/24 h had no significant difference compared with dry aging and wet aging. The MFI of the groups with moisture permeabilities of 7900 and 11,000 g/m^2^/24 h was significantly higher than that of wet aging after 14 d (*p* < 0.05), and there was no significant difference compared with dry aging. Dry aging group had the highest MFI values among treatments aged for 7 and 14 d. MFI had a decreasing tendency with decreasing moisture permeability. Therefore, the moisture-permeable packaging could effectively influence the MFI of beef. This was consistent with the results of shear force and TPA as shown above. Shin [23] reported that shear force had a negative correlation with MFI. The decrease in shear force and hardness may be due to the increase in MFI.

### 3.7. Low-Field NMR Analysis

Low-field NMR is used to study the water distribution and mobility of beef. There were three different peaks found in the inversion map of nuclear magnetic intensity. According to the relaxation time (T_2_), three types of water were divided into bound water (T_2b_, 1–10 ms), capillary water (T_21_, 10–100 ms), and free water (T_22_, 100–1000 ms) in Figure 1. As shown in Table 6, aging time had a significant effect on the peak area percentage of the bound water (P_2b_) in beef (*p* < 0.05). There was no significant difference among groups except the group with moisture permeabilities of 350 g/m^2^/24 h. P_2b_ for the dry aging group was significantly lower than that of other treatments after 14 d (*p* < 0.05). According to the results of capillary water, the aging time, aging methods, and their interaction all had significant effects on the percentage of capillary water (P_21_). There was no significant difference among groups except the group with moisture permeabilities of 350 g/m^2^/24 h after 7 d. The P_21_ was the highest in dry-aged samples that were aged for 14 d, followed by the moisture-permeable packaging treatments, and the lowest P_21_ was in wet-aged samples. According to the free water analysis, there was no significant difference between the six groups except the 350 group. The percentage of free water (P_22_) in dry-aged samples was significantly lower than that of other groups after 14 d (*p* < 0.05). P_2b_ in dry-aged samples was the lowest among the groups, possibly due to the protein denaturation and myofibril structure changes, leading to more bound water being released. Because dry aging had the highest MFI, it indicated that the number of hydrogen bonds in the interaction between proteins, ions, and water molecules was reduced, resulting in a decrease of P_2b_ [24]. Moreover, the released bound water was transformed into capillary water and free water. In addition, the moisture in the beef was evaporated due to the moisture permeability of moisture-permeable packaging, and the free water was the first to evaporate. The P_22_ of dry aging and the groups having high moisture permeability was relatively low. Therefore, different moisture-permeable packagings could have a significant influence on the distribution of water in beef.

## 4. Conclusions

The data from our study showed that moisture-permeable packaging was able to effectively control weight loss. Microbial population of samples aging in moisture-permeable packaging was inhabited compared with dry aging. Additionally, different moisture-permeable packaging affected apparently shear force and texture of beef. The decrease of MFI with the increase of moisture permeability indicated that it appeared to cause different damage in the myofibril of beef. The damage of myofibril and evaporation of water resulted in changes in the distribution of water. Thus, these attributes can be controlled and affected by implementing different moisture-permeable packagings to control the evaporation of water. 

## Figures and Tables

**Figure 1 foods-09-00649-f001:**
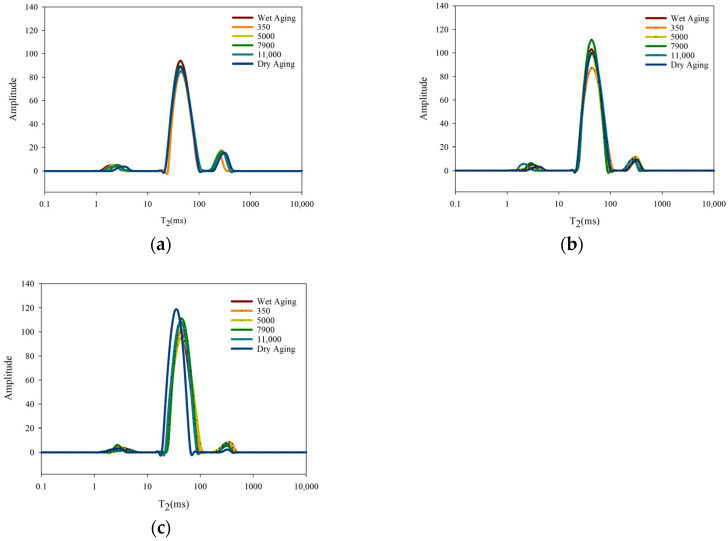
(**a**) low-field NMR T_2_ relaxation curves of aged beef with different treatments (0 d); (**b**) low-field NMR T_2_ relaxation curves of aged beef with different treatments (7 d); (**c**) low-field NMR T_2_ relaxation curves of aged beef with different treatments (14 d).

**Table 1 foods-09-00649-t001:** Effects of different aging methods and aging time on the weight loss (%).

Trait	Aging Time (d)	Aging Methods ^1^						SE ^3^	*p*-Values ^2^		
		Wet aging	350	5000	7900	11,000	Dry Aging		Time	Method	Time × Method
Aging Loss	0	- ^4^	-	-	-	-	-	0.34	<0.001	<0.001	<0.001
	7	1.27 ^cx 5^	1.28 ^cy^	1.30 ^cy^	1.95 ^by^	1.83 ^by^	3.13 ^ay^				
	14	1.44 ^dx^	1.95 ^cx^	2.12 ^cx^	6.16 ^bx^	7.66 ^ax^	7.46 ^ax^				
Trim Loss	0	-	-	-	-	-	-	1.02	<0.001	<0.001	0.074
	7	-	-	-	4.28 ^cy^	7.22 ^by^	12.43 ^ay^				
	14	-	5.72 ^cx^	7.37 ^cx^	10.11 ^bx^	13.87 ^bx^	21.73 ^ax^				
Total Loss	0	15.34 ^ax^	14.82 ^ay^	16.80 ^ax^	17.88 ^ay^	14.19 ^az^	16.01 ^az^	0.97	<0.001	<0.001	<0.001
	7	17.01 ^cx^	15.30 ^cy^	18.60 ^bcx^	20.04 ^bcy^	22.37 ^by^	28.93 ^ay^				
	14	15.00 ^dx^	24.15 ^cx^	22.73 ^cdx^	29.66 ^bcx^	34.21 ^abx^	39.95 ^ax^				

^1^ Under aging methods, 350, 5000, 7900, and 11,000 mean the transmission of water vapor of four different moisture-permeable packagings, respectively; ^2^
*p*-values after columns indicate the level of significance including aging time, aging methods, and their interaction; ^3^ SE: standard error; ^4^ -: indicates loss did not exist. ^5^ a~d within the same row indicates a significant difference (*p* < 0.05) between aging methods, x~z within the same column indicates a significant difference (*p* < 0.05) between aging time.

**Table 2 foods-09-00649-t002:** Effects of different aging methods and aging time on the pH, L*, a*, b*, cooking loss, and water content of beef.

Trait	Aging Time (d)	Aging Methods ^1^						SE ^3^	*p*-Values ^2^		
		Wet Aging	350	5000	7900	11,000	Dry Aging		Time	Method	Time × Method
pH	0	5.40 ^ay^ ^4^	5.40 ^ay^	5.37 ^ay^	5.38 ^ay^	5.41 ^ay^	5.41 ^ay^	0.01	<0.001	0.848	0.466
	7	5.57 ^ax^	5.58 ^ax^	5.58 ^ax^	5.59 ^ax^	5.58 ^ax^	5.58 ^ax^				
	14	5.63 ^ax^	5.59 ^ax^	5.62 ^ax^	5.59 ^ax^	5.61 ^ax^	5.61 ^ax^				
L*	0	37.75 ^ax^	37.58 ^ax^	37.59 ^ax^	37.12 ^ax^	37.56 ^ax^	38.25 ^ax^	0.15	0.179	0.001	0.007
	7	38.39 ^abx^	37.63 ^abcx^	39.26 ^ax^	34.96 ^dy^	37.14 ^bcx^	35.92 ^cdy^				
	14	38.29 ^ax^	38.25 ^ax^	38.26 ^ax^	37.78 ^ax^	37.71 ^ax^	36.69 ^by^				
a*	0	17.31 ^ax^	16.43 ^ax^	17.20 ^ax^	16.53 ^ax^	16.05 ^ax^	17.36 ^ax^	0.13	<0.001	0.382	0.198
	7	15.2 ^aby^	16.26 ^ax^	14.51 ^by^	15.57 ^abx^	15.16 ^abx^	15.8 ^aby^				
	14	14.78 ^ay^	15.76 ^ax^	15.85 ^ay^	14.87 ^ax^	14.57 ^ax^	14.81 ^ay^				
b*	0	6.29 ^ax^	6.49 ^ax^	7.23 ^ax^	5.95 ^ax^	6.61 ^ax^	7.60 ^ax^	0.12	<0.001	0.003	0.004
	7	6.19 ^ax^	6.33 ^ax^	6.09 ^ay^	6.17 ^ax^	6.11 ^ax^	7.03 ^ax^				
	14	5.24 ^abx^	7.3 ^ax^	7.05 ^axy^	4.62 ^bx^	4.81 ^by^	4.67 ^by^				
Cooking Loss (%)	0	23.09 ^ay^	23.28 ^ay^	22.66 ^ay^	23.66 ^ay^	25.00 ^ax^	25.04 ^ay^	0.28	<0.001	0.001	0.001
	7	25.79 ^abcx^	27.97 ^abx^	28.27 ^ax^	23.51 ^cy^	25.10 ^bcx^	25.09 ^bcy^				
	14	21.27 ^cy^	28.82 ^ax^	28.22 ^abx^	25.90 ^bx^	25.64 ^bx^	27.41 ^abx^				
Water Content (%)	0	73.35 ^ax^	73.74 ^ax^	73.67 ^ax^	73.54 ^ax^	73.32 ^ax^	73.98 ^ax^	0.11	0.102	0.188	0.730
	7	73.92 ^ax^	73.49 ^ax^	73.83 ^ax^	73.67 ^ax^	73.76 ^ax^	72.49 ^by^				
	14	73.31 ^ax^	73.92 ^ax^	73.27 ^ax^	73.84 ^ax^	73.64 ^ax^	72.90 ^by^				

^1^ Under aging methods, 350, 5000, 7900, and 11,000 mean the transmission of water vapor of four different moisture-permeable packagings, respectively; ^2^
*p*-values after columns indicate the level of significance including aging time, aging methods, and their interaction; ^3^ SE: standard error; ^4^ a~d within the same row indicates a significant difference (*p* < 0.05) between aging methods, x~z within the same column indicates a significant difference (*p* < 0.05) between aging time.

**Table 3 foods-09-00649-t003:** Effects of different aging methods and aging time on the shear force and texture profile analysis (TPA) of beef.

Trait	Aging Time (d)	Aging Methods ^1^						SE ^3^	*p*-Values ^2^		
		Wet Aging	350	5000	7900	11,000	Dry Aging		Time	Method	Time × Method
Shear Force (N)	0	81.13 ^ax^ ^4^	76.42 ^az^	68.08 ^ax^	76.62 ^ax^	79.36 ^ax^	81.62 ^ax^	2.58	<0.001	0.199	0.006
	7	29.82 ^bcy^	43.56 ^ay^	35.81 ^aby^	30.71 ^bcy^	32.86 ^bcy^	25.70 ^cy^				
	14	33.84 ^ay^	31.59 ^ax^	33.84 ^ay^	27.66 ^aby^	29.04 ^ay^	22.79 ^by^				
Hardness	0	13,885.00 ^ax^	15,676.31 ^ax^	14,344.76 ^ax^	15,036.12 ^ax^	14,462.26 ^ax^	15,988.77 ^ax^	203	<0.001	<0.001	<0.001
	7	14,489.43 ^ax^	14,006.18 ^ax^	13,497.55 ^ax^	10,995.27 ^by^	11,386.99 ^by^	11,136.72 ^by^				
	14	13,711.00 ^abx^	14,911.28 ^ax^	14,359.48 ^ax^	11,657.09 ^by^	11,386.99 ^by^	9133.81 ^cz^				
Springiness	0	0.464 ^ay^	0.463 ^ax^	0.442 ^ay^	0.465 ^ay^	0.479 ^ax^	0.475 ^ax^	0.01	0.001	0.468	0.667
	7	0.518 ^ax^	0.482 ^ax^	0.518 ^ax^	0.503 ^axy^	0.499 ^ax^	0.507 ^ax^				
	14	0.475 ^axy^	0.470 ^ax^	0.476 ^axy^	0.514 ^ax^	0.502 ^ax^	0.495 ^ax^				
Cohesiveness	0	0.519 ^axy^	0.493 ^ax^	0.480 ^ay^	0.519 ^ax^	0.503 ^ax^	0.517 ^ax^	0.01	<0.001	0.056	0.003
	7	0.533 ^ax^	0.490 ^abx^	0.530 ^ax^	0.484 ^by^	0.490 ^bx^	0.484 ^bx^				
	14	0.487 ^ay^	0.481 ^ax^	0.506 ^axy^	0.478 ^ay^	0.481 ^ax^	0.441 ^by^				
Chewiness	0	3145.06 ^ax^	3768.84 ^ax^	3181.66 ^ax^	3703.85 ^ax^	3646.82 ^ax^	3947.95 ^ax^	74.1	<0.001	0.08	0.003
	7	3878.69 ^ax^	3333.02 ^abx^	3809.51 ^ax^	2582.78 ^by^	2859.62 ^by^	2729.91 ^by^				
	14	3022.42 ^abx^	3222.11 ^abx^	3456.25 ^ax^	2825.88 ^by^	2760.79 ^by^	2035.61 ^cy^				

^1^ Under aging methods, 350, 5000, 7900, and 11,000 mean the transmission of water vapor of four different moisture-permeable packagings, respectively; ^2^
*p*-values after columns indicate the level of significance including aging time, aging methods, and their interaction; ^3^ SE: standard error; ^4^ a~d within the same row indicates a significant difference (*p* < 0.05) between aging methods, x~z within the same column indicates a significant difference (*p* < 0.05) between aging time.

**Table 4 foods-09-00649-t004:** Effects of different aging methods and aging time on the microbial population (log colony-forming unit (cfu)/g meat) of beef.

Trait	Aging Time (d)	Aging Methods ^1^						SE ^3^	*p*-Values ^2^		
		Wet Aging	350	5000	7900	11,000	Dry Aging		Time	Method	Time × Method
Aerobic Bacterial Count	0	4.68 ^az^ ^4^	4.85 ^ay^	4.72 ^az^	4.59 ^ay^	4.83 ^ay^	4.62 ^az^	0.09	<0.001	<0.001	<0.001
	7	5.97 ^cy^	6.57 ^bx^	5.53 ^dy^	6.72 ^bx^	6.60 ^bx^	7.21 ^ay^				
	14	6.27 ^cx^	6.50 ^bcx^	6.71 ^bx^	6.67 ^bx^	6.83 ^bx^	7.53 ^ax^				
Lactic Acid Bacteria	0	3.54 ^az^	3.66 ^az^	3.47 ^az^	3.82 ^az^	3.51 ^az^	3.59 ^az^	0.08	<0.001	0.349	<0.001
	7	5.14 ^by^	5.88 ^ax^	5.84 ^ax^	5.56 ^ax^	5.85 ^ax^	5.73 ^ax^				
	14	5.42 ^ax^	4.65 ^dy^	5.08 ^bcy^	4.99 ^cy^	5.03 ^cy^	5.29 ^aby^				
Mold	0	- ^5^	-	-	-	-	-	0.07	<0.001	<0.001	<0.001
	7	1.38 ^az^	1.67 ^ay^	1.39 ^az^	1.43 ^az^	1.38 ^az^	1.34 ^az^				
	14	1.98 ^cy^	1.48 ^dx^	1.89 ^cy^	2.13 ^bcy^	2.36 ^by^	3.14 ^ay^				
Yeast	0	2.59 ^az^	2.71 ^ay^	2.41 ^az^	2.42 ^az^	2.55 ^ay^	2.53 ^az^	0.09	<0.001	<0.001	<0.001
	7	5.24 ^cx^	5.43 ^bcx^	5.36 ^bcx^	5.49 ^bx^	5.49 ^bx^	5.74 ^ay^				
	14	3.99 ^cy^	3.28 ^dy^	4.17 ^cy^	5.11 ^by^	5.46 ^bx^	6.12 ^ax^				

^1^ Under aging methods, 350, 5000, 7900, and 11,000 mean the transmission of water vapor of four different moisture-permeable packagings, respectively; ^2^
*p*-values after columns indicate the level of significance including aging time, aging methods, and their interaction; ^3^ SE: standard error; ^4^ a~d within the same row indicates a significant difference (*p* < 0.05) between aging methods, x~z within the same column indicates a significant difference (*p* < 0.05) between aging time; ^5^ -: no detection.

**Table 5 foods-09-00649-t005:** Effects of different aging methods and aging time on the myofibrillar fragmentation index (MFI) of beef.

Trait	Aging Time(d)	Aging Methods ^1^						SE ^3^	*p*-Values ^2^		
		Wet Aging	350	5000	7900	11,000	Dry Aging		Time	Method	Time × Method
MFI	0	16.99 ^az^	16.21 ^az^	15.78 ^az^	17.52 ^az^	16.12 ^az^	15.72 ^az^	0.30	<0.001	<0.001	0.001
	7	35.22 ^aby^	29.31 ^cy^	32.48 ^bcy^	37.02 ^aby^	36.5 ^aby^	39.26 ^ay^				
	14	44.85 ^cx^	43.99 ^cx^	48.79 ^bcx^	54.96 ^abx^	52.51 ^abx^	53.73 ^ax^				

^1^ Under aging methods, 350, 5000, 7900, and 11,000 mean the transmission of water vapor of four different moisture-permeable packagings, respectively; ^2^
*p*-values after columns indicate the level of significance including aging time, aging methods, and their interaction; ^3^ SE: standard error; a~d within the same row indicates a significant difference (*p* < 0.05) between aging methods, x~z within the same column indicates a significant difference (*p* < 0.05) between aging time; SF: surface hydrophobicity.

**Table 6 foods-09-00649-t006:** Effects of different aging methods and different aging time on the moisture distribution of beef.

Trait	Aging Time (d)	Aging Methods ^1^						SE ^3^	*p*-Values ^2^		
		Wet Aging	350	5000	7900	11,000	Dry Aging		Time	Method	Time × Method
P_2b_	0	4.28 ^ax^	4.66 ^ax^	3.83 ^ax^	4.80 ^ax^	4.38 ^ax^	4.01 ^ax^	0.10	<0.001	0.145	0.571
	7	3.64 ^bx^	4.49 ^ax^	3.49 ^bx^	3.43 ^bx^	3.39 ^by^	3.4 ^bxy^				
	14	3.67 ^ax^	3.44 ^aby^	3.76 ^ax^	3.38 ^abx^	3.36 ^aby^	2.73 ^by^				
P_21_	0	87.73 ^ay^	87.77 ^ay^	88.33 ^ay^	86.92 ^ay^	86.55 ^az^	87.46 ^ay^	0.36	<0.001	0.011	0.009
	7	92.03 ^ax^	88.14 ^by^	91.48 ^ax^	93.54 ^ax^	91.97 ^ay^	92.43 ^ax^				
	14	91.39 ^cx^	91.91 ^bcx^	92.46 ^bcx^	93.6 ^bx^	93.71 ^bx^	95.63 ^ax^				
P_22_	0	7.99 ^ax^	7.57 ^ax^	7.84 ^ax^	8.28 ^ax^	9.06 ^ax^	8.53 ^ax^	0.30	<0.001	0.182	0.124
	7	4.33 ^by^	7.37 ^ax^	5.03 ^by^	3.43 ^by^	4.64 ^by^	4.18 ^by^				
	14	4.94 ^ay^	4.64 ^ay^	3.79 ^ay^	3.02 ^by^	3.18 ^bz^	2.34 ^cz^				

^1^ Under aging methods, 350, 5000, 7900, and 11,000 mean the transmission of water vapor of four different moisture-permeable packagings, respectively; ^2^
*p*-values after columns indicate the level of significance including aging time, aging methods, and their interaction; ^3^ SE: standard error; a~d within the same row indicates a significant difference (*p* < 0.05) between aging methods, x~z within the same column indicates a significant difference (*p* < 0.05) between aging time.

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
