# Peer review of "Effects of Different Moisture-Permeable Packaging on the Quality of Aging Beef Compared with Wet Aging and Dry Aging"

_foods, 2020, doi:10.3390/foods9050649_

Round 1

Reviewer 1 Report

Summary:

Although it is not explicitly described in the abstract or introduction, the overall objective of this study was to investigate the impact of six different aging methods and three aging times on the physical, chemical, microbial and sensory quality of beef. The study reported that aging beef in moisture permeable bags could effectively reduce weight loss and microbial growth while subsequently improve sensory quality in terms of tenderness and juiciness. The paper concludes that "different moisture permeable bags can effect weight, shear force, color and microbial counts, and thus, these attributes are able to be controlled via moisture permeable bags controlling evaporation of water".

Overall comment:

It is clear from the introduction, and the reviewers own knowledge of the subject, that a considerable amount of literature already exists examining the impact of different aging methods and aging times on various aspects of beef quality. The authors argue that while previous research studies have been conducted on this topic, the results to date have been contradictory, particularly in terms of the differences found between aging methods on sensory perception across studies. Therefore, following the introduction, I assumed that sensory analysis would form a significant component of this research study, and contribute to the novelty of the work.

However, the sample size used in the sensory consumer study (n=12) is extremely small and no valid statistical conclusions can be drawn from these sensory results. A sample size of 12 participants is more typical of studies using descriptive sensory analysis with a trained sensory panel. Using so few consumers to generate hedonic data is not appropriate in my opinion.

I can therefore not recommend acceptance of the paper as it is. If it is not possible to recruit additional consumers to participate in the sensory tests, I recommend that the authors rearrange the manuscript focusing on the physicochemical properties and microbial data as an indication of beef quality. The results from the consumer sensory testing should only be discussed within the context of the study.

Reviewer 2 Report

Foods Review of:

Shi et al., “Effects of different moisture permeable package on the quality of aging beef compared with wet aging and dry aging.”

Overall:  It is the professional opinion of the reviewer that the study appears to be a valid and important contribution to the body of literature regarding dry-aging of beef products.  However, because of the immense amounts of grammatical errors and incomplete written thoughts, I am unable to complete the review of this paper.  It is the recommendation of the reviewer to the authors to work on grammatical sentence structure in order to produce a document that will read easier for the audience in question.  Unfortunately, it is the reviewer’s professional opinion that this paper is not ready for publication.

Some Additional Comments:

Ln 80: Be precise in number of days aged/stored.  Months are variable in the number of days.

Ln 89 – 91: The Minolta CR400 is not a pH meter but rather a color spectrophotometer.  Please describe the actual means of measuring pH.  The Minolta Cr-40 is not a spectrophotometer.  Please review what instruments were used and in what manner.

Ln 97-98.  This is a rather unorthodox means of conducting Warner-Bratzler shear force analyses.  Please provide precedence of this unusual method or provide why this unusual method was used.

Ln 140: Why were the days of aging used?  Commercial dry-aging facilities generally age for more than 21 days.

Tables: Provide better descriptions of significance notations so that the reader knows what is being compared.

Round 2

Reviewer 1 Report

Overall comment:

The removal of sensory data in place of the MFI and Low-field NMR analysis improves the novelty of the paper. However, as per my original recommendation, this paper requires extensive editing in terms of English language, and therefore I cannot review, or recommend the paper, as present. I recommend that the authors carefully review their paper, or consider editing services, to correct grammatical errors, and to conform to correct scientific English and interpretation. 

Reviewer 2 Report

Foods Review of:

Shi et al., “Effects of different moisture permeable package on the quality of aging beef compared with wet aging and dry aging.”

Overall:  It is the professional opinion of the reviewer that the study appears to be a valid and important contribution to the body of literature regarding dry-aging of beef products.  However, because of the immense amounts of grammatical errors and incomplete written thoughts, I am still unable to complete the review of this paper.  It is the recommendation of the reviewer to the authors to work on grammatical sentence structure in order to produce a document that will read easier for the audience in question.  Unfortunately, it is the reviewer’s professional opinion that this paper is not ready for publication.

Some Additional Comments:

Ln 2 – 4: Work on the title so that it reads more grammatically correct.  Example: “Effects of different moisture-permeable packaging/packages/packaging systems…”

Ln 15 – 16: how did yeast and mold increase? Relative abundance of species and/or number of colony forming units?

Ln 17: Explain “slight decrease in color”.

Ln 18 – 20: Incomplete thoughts and sentences.  P2b and P22 not defined.  MFI does not need abbreviating since it is only used one additional time in the abstract.

Ln 22: Was there an economic assessment conducted in the study?  Dry aging is often considered a value-added step and although there are greater yield losses, those dry-aging are able to ask for higher per unit money values thus still making it a financially viable means of merchandising meat.

Ln 12 – 24: Abstract needs to be proof-read prior to submission.  Many grammatical errors and incomplete thoughts. 

The reviewer was unwilling to continue to write all of the additional grammatical and sentence structure revisions necessary for this document to be accepted.  Please critically proof-read and review the paper prior to resubmission.  Look for sentences that are incomplete.  Check formatting guides for proper document formatting. The science seems sound, but because of how it is currently written I cannot complete the review.
